# Unsupervised Representation Learning by Predicting Random Distances

## Abstract

Deep neural networks have gained tremendous success in a broad range of machine learning tasks due to its remarkable capability to learn semantic-rich features from high-dimensional data. However, they often require large-scale labelled data to successfully learn such features, which significantly hinders their adaption into unsupervised learning tasks, such as anomaly detection and clustering, and limits their applications into critical domains where obtaining massive labelled data is prohibitively expensive. To enable downstream unsupervised learning on those domains, in this work we propose to learn features without using any labelled data by training neural networks to predict data distances in a randomly projected space. Random mapping is a theoretical proven approach to obtain approximately preserved distances. To well predict these random distances, the representation learner is optimised to learn genuine class structures that are implicitly embedded in the randomly projected space. Experimental results on 19 real-world datasets show our learned representations substantially outperform state-of-the-art competing methods in both anomaly detection and clustering tasks.

## 1 Introduction

Unsupervised representation learning aims at automatically extracting expressive feature representations from data without any manually labelled data. Due to the remarkable capability to learn semantic-rich features, deep neural networks have been becoming one widely-used technique to empower a broad range of machine learning tasks. One main issue with these deep learning techniques is that a massive amount of labelled data is typically required to successfully learn these expressive features. As a result, their transformation power is largely reduced for tasks that are unsupervised in nature, such as anomaly detection and clustering. This is also true to critical domains, such as healthcare and fintech, where collecting massive labelled data is prohibitively expensive and/or is impossible to scale. To bridge this gap, in this work we explore fully unsupervised representation learning techniques to enable downstream unsupervised learning methods on those critical domains.

In recent years, many unsupervised representation learning methods (Mikolov et al., 2013a; Le & Mikolov, 2014; Misra et al., 2016; Lee et al., 2017; Gidaris et al., 2018) have been introduced, of which most are self-supervised approaches that formulate the problem as an annotation free pretext task. These methods explore easily accessible information, such as temporal or spatial neighbourhood, to design a surrogate supervisory signal to empower the feature learning. These methods have achieved significantly improved feature representations of text/image/video data, but they are often inapplicable to *tabular data* since it does not contain the required temporal or spatial supervisory information. We therefore focus on unsupervised representation learning of high-dimensional tabular data. Although many traditional approaches, such as random projection (Li et al., 2006), principal component analysis (PCA) (Rahmani & Atia, 2017), manifold learning (Donoho & Grimes, 2003; Hinton & Roweis, 2003) and autoencoder (Vincent et al., 2010), are readily available for handling those data, many of them (Donoho & Grimes, 2003; Hinton & Roweis, 2003; Rahmani & Atia, 2017) are often too computationally costly to scale up to large or high-dimensional data. Approaches like random projection and autoencoder are very efficient but they often fail to capture complex class structures due to its underlying data assumption or weak supervisory signal.

In this paper, we introduce a Random Distance Prediction (RDP) model which trains neural networks to predict data distances in a randomly projected space. When the distance information captures in-

trinsic class structure in the data, the representation learner is optimised to learn the class structure to minimise the prediction error. Since distances are concentrated and become meaningless in high dimensional spaces (Beyer et al., 1999), we seek to obtain distances preserved in a projected space to be the supervisory signal. Random mapping is a highly efficient yet theoretical proven approach to obtain such approximately preserved distances. Therefore, we leverage the distances in the randomly projected space to learn the desired features. Intuitively, random mapping preserves rich local proximity information but may also keep misleading proximity when its underlying data distribution assumption is inexact; by minimising the random distance prediction error, RDP essentially leverages the preserved data proximity and the power of neural networks to learn globally consistent proximity and rectify the inconsistent proximity information, resulting in a substantially better representation space than the original space. We show this simple random distance prediction enables us to achieve expressive representations with no manually labelled data. In addition, some task-dependent auxiliary losses can be optionally added as a complementary supervisory source to the random distance prediction, so as to learn the feature representations that are more tailored for a specific downstream task. In summary, this paper makes the following three main contributions.

- We propose a random distance prediction formulation, which is very simple yet offers a highly effective supervisory signal for learning expressive feature representations that *optimise* the distance preserving in random projection. The learned features are sufficiently generic and work well in enabling different downstream learning tasks.
- Our formulation is flexible to incorporate task-dependent auxiliary losses that are complementary to random distance prediction to further enhance the learned features, i.e., features that are specifically optimised for a downstream task while at the same time preserving the generic proximity as much as possible.
- As a result, we show that our instantiated model termed RDP enables substantially better performance than state-of-the-art competing methods in two key unsupervised tasks, anomaly detection and clustering, on 19 real-world high-dimensional tabular datasets.

## 2 RANDOM DISTANCE PREDICTION MODEL

### 2.1 THE PROPOSED FORMULATION AND THE INSTANTIATED MODEL

We propose to learn representations by training neural networks to predict distances in a randomly projected space without manually labelled data. The key intuition is that, given some distance information that faithfully encapsulates the underlying class structure in the data, the representation learner is forced to learn the class structure in order to yield distances that are as close as the given distances. Our proposed framework is illustrated in Figure 1. Specifically, given data points $\mathbf{x}_i, \mathbf{x}_j \in \mathbb{R}^D$, we first feed them into a weight-shared Siamese-style neural network $\phi(\mathbf{x}; \Theta)$. $\phi : \mathbb{R}^D \mapsto \mathbb{R}^M$ is a representation learner with the parameters $\Theta$ to map the data onto a $M$-dimensional new space. Then we formulate the subsequent step as a distance prediction task and define a loss function as:

$$L_{rdp}(\mathbf{x}_i, \mathbf{x}_j) = l(\langle \phi(\mathbf{x}_i; \Theta), \phi(\mathbf{x}_j; \Theta) \rangle, \langle \eta(\mathbf{x}_i), \eta(\mathbf{x}_j) \rangle), \qquad (1)$$

where $\eta$ is an existing projection method and $l$ is a function of the difference between its two inputs.

Here one key ingredient is how to obtain trustworthy distances via $\eta$. Also, to efficiently optimise the model, the distance derivation needs to be computationally efficient. In this work, we use the inner products in a randomly projected space as the source of distance/similarity since it is very efficient and there is strong theoretical support of its capacity in preserving the genuine distance information. Thus, our instantiated model RDP specifies $L_{rdp}(\mathbf{x}_i, \mathbf{x}_j)$ as follows[1]:

$$L_{rdp}(\mathbf{x}_i, \mathbf{x}_j) = (\phi(\mathbf{x}_i; \Theta) \cdot \phi(\mathbf{x}_j; \Theta) - \eta(\mathbf{x}_i) \cdot \eta(\mathbf{x}_j))^2, \qquad (2)$$

where $\phi$ is implemented by multilayer perceptron for dealing with tabular data and $\eta : \mathbb{R}^D \mapsto \mathbb{R}^K$ is an off-the-shelf random data mapping function (see Sections 3.1 and 3.2 for detail). Despite its simplicity, this loss offers a powerful supervisory signal to learn semantic-rich feature representations that substantially optimise the underlying distance preserving in $\eta$ (see Section 3.3 for detail).

---

[1]Since we operate on real-valued vector space, the inner product is implemented by the dot product. The dot product is used hereafter to simplify the notation.

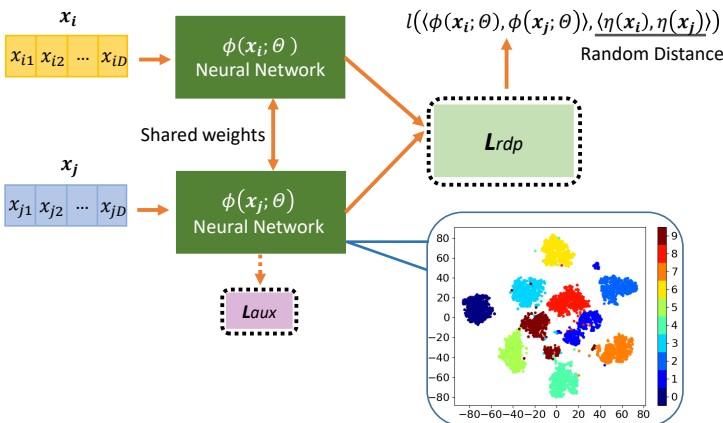

Figure 1: The proposed random distance prediction (RDP) framework. Specifically, a weight-shared two-branch neural network $\phi$ first projects $\mathbf{x}_i$ and $\mathbf{x}_j$ onto a new space, in which we aim to minimise the random distance prediction loss $L_{rdp}$, i.e., the difference between the learned distance $\langle \phi(\mathbf{x}_i; \Theta), \phi(\mathbf{x}_j; \Theta) \rangle$ and a predefined distance $\langle \eta(\mathbf{x}_i), \eta(\mathbf{x}_j) \rangle$ ($\eta$ denotes an existing random mapping). $L_{aux}$ is an auxiliary loss that is optionally applied to one network branch to learn complementary information w.r.t. $L_{rdp}$. The lower right figure presents a 2-D t-SNE (Hinton & Roweis, 2003) visualisation of the features learned by RDP on a small toy dataset *optdigits* with 10 classes.

## 2.2 FLEXIBILITY TO INCORPORATE TASK-DEPENDENT COMPLEMENTARY AUXILIARY LOSS

Minimising $L_{rdp}$ learns to preserve pairwise distances that are critical to different learning tasks. Moreover, our formulation is flexible to incorporate a task-dependent auxiliary loss $L_{aux}$, such as reconstruction loss (Hinton & Salakhutdinov, 2006) for clustering or novelty loss (Burda et al., 2019) for anomaly detection, to complement the proximity information and enhance the feature learning.

For clustering, an auxiliary reconstruction loss is defined as:

$$L_{aux}^{clu}(\mathbf{x}) = (\mathbf{x} - \phi'(\phi(\mathbf{x}; \Theta); \Theta'))^2, \tag{3}$$

where $\phi$ is an encoder and $\phi' : \mathbb{R}^M \mapsto \mathbb{R}^D$ is a decoder. This loss may be optionally added into RDP to better capture global feature representations.

Similarly, in anomaly detection a novelty loss may be optionally added, which is defined as:

$$L_{aux}^{ad}(\mathbf{x}) = (\phi(\mathbf{x}; \Theta) - \eta(\mathbf{x}))^2. \tag{4}$$

By using a fixed $\eta$, minimising $L_{aux}^{ad}$ helps learn the frequency of underlying patterns in the data (Burda et al., 2019), which is an important complementary supervisory source for the sake of anomaly detection. As a result, anomalies or novel points are expected to have substantially larger $(\phi(\mathbf{x}; \Theta^\star) - \eta(\mathbf{x}))^2$ than normal points, so this value can be directly leveraged to detect anomalies.

Note since $L_{aux}^{ad}$ involves a mean squared error between two vectors, the dimension of the projected space resulted by $\phi$ and $\eta$ is required to be equal in this case. Therefore, when this loss is added into RDP, the $M$ in $\phi$ and $K$ in $\eta$ need to be the same. We do not have this constraint in other cases.

## 3 THEORETICAL ANALYSIS OF RDP

This section shows the proximity information can be well approximated using inner products in two types of random projection spaces. This is a key theoretical foundation to RDP. Also, to accurately predict these distances, RDP is forced to learn the genuine class structure in the data.

### 3.1 WHEN LINEAR PROJECTION IS USED

Random projection is a simple yet very effective linear feature mapping technique which has proven the capability of distance preservation. Let $\mathcal{X} \subset \mathbb{R}^{N \times D}$ be a set of $N$ data points, random projection

uses a random matrix $\mathbf{A} \subset \mathbb{R}^{K \times D}$ to project the data onto a lower $K$-dimensional space by $\mathcal{X}' = \mathbf{A}\mathcal{X}^\mathsf{T}$. The Johnson-Lindenstrauss lemma (Johnson & Lindenstrauss, 1984) guarantees the data points can be mapped to a randomly selected space of suitably lower dimension with the distances between the points are approximately preserved. More specifically, let $\epsilon \in (0, \frac{1}{2})$ and $K = \frac{20 \log n}{\epsilon^2}$. There exists a linear mapping $f : \mathbb{R}^D \mapsto \mathbb{R}^K$ such that for all $\mathbf{x}_i, \mathbf{x}_j \in \mathcal{X}$:

$$(1 - \epsilon)||\mathbf{x}_i - \mathbf{x}_j||^2 \leq ||f(\mathbf{x}_i) - f(\mathbf{x}_j)||^2 \leq (1 + \epsilon)||\mathbf{x}_i - \mathbf{x}_j||^2. \tag{5}$$

Furthermore, assume the entries of the matrix $\mathbf{A}$ are sampled independently from a Gaussian distribution $\mathcal{N}(0, 1)$. Then, the norm of $\mathbf{x} \in \mathbb{R}^D$ can be preserved as:

$$\Pr\left((1 - \epsilon)||\mathbf{x}||^2 \leq ||\frac{1}{\sqrt{K}}\mathbf{A}\mathbf{x}||^2 \leq (1 + \epsilon)||\mathbf{x}||^2\right) \geq 1 - 2e^{\frac{-(\epsilon^2 - \epsilon^3)K}{4}}. \tag{6}$$

Under such random projections, the norm preservation helps well preserve the inner products:

$$\Pr\left(|\hat{\mathbf{x}}_i \cdot \hat{\mathbf{x}}_j - f(\hat{\mathbf{x}}_i) \cdot f(\hat{\mathbf{x}}_j)| \geq \epsilon\right) \leq 4e^{\frac{-(\epsilon^2 - \epsilon^3)K}{4}}, \tag{7}$$

where $\hat{\mathbf{x}}$ is a normalised $\mathbf{x}$ such that $||\hat{\mathbf{x}}|| \leq 1$.

The proofs of Eqns. (5), (6) and (7) can be found in (Vempala, 1998).

Eqn. (7) states that the inner products in the randomly projected space can largely preserve the inner products in the original space, particularly when the projected dimension $K$ is large.

## 3.2 When Non-linear Projection Is Used

Here we show that some non-linear random mapping methods are approximate to kernel functions which are a well-established approach to obtain reliable distance/similarity information. The key to this approach is the kernel function $k : \mathcal{X} \times \mathcal{X} \mapsto \mathbb{R}$, which is defined as $k(\mathbf{x}_i, \mathbf{x}_j) = \langle \psi(\mathbf{x}_i), \psi(\mathbf{x}_j) \rangle$, where $\psi$ is a feature mapping function but needs not to be explicitly defined and $\langle \cdot, \cdot \rangle$ denotes a suitable inner product. A non-linear kernel function such as polynomial or radial basis function (RBF) kernel is typically used to project linear-inseparable data onto a linear-separable space.

The relation between non-linear random mapping and kernel methods is justified in (Rahimi & Recht, 2008), which shows that an explicit randomised mapping function $g : \mathbb{R}^D \mapsto \mathbb{R}^K$ can be defined to project the data points onto a low-dimensional Euclidean inner product space such that the inner products in the projected space approximate the kernel evaluation:

$$k(\mathbf{x}_i, \mathbf{x}_j) = \langle \psi(\mathbf{x}_i), \psi(\mathbf{x}_j) \rangle \approx g(\mathbf{x}_i) \cdot g(\mathbf{x}_j). \tag{8}$$

Let $\mathbf{A}$ be the mapping matrix. Then to achieve the above approximation, $\mathbf{A}$ is required to be drawn from Fourier transform and shift-invariant functions such as cosine function are finally applied to $\mathbf{A}\mathbf{x}$ to yield a real-valued output. By transforming the two data points $\mathbf{x}_i$ and $\mathbf{x}_j$ in this manner, their inner product $g(\mathbf{x}_i) \cdot g(\mathbf{x}_j)$ is an unbiased estimator of $k(\mathbf{x}_i, \mathbf{x}_j)$.

## 3.3 Learning Class Structure By Random Distance Prediction

Our model using only the random distances as the supervisory signal can be formulated as:

$$\underset{\Theta}{\arg\min} \sum_{\mathbf{x}_i, \mathbf{x}_j \in \mathcal{X}} \left(\phi(\mathbf{x}_i; \Theta) \cdot \phi(\mathbf{x}_j; \Theta) - y_{ij}\right)^2, \tag{9}$$

where $y_{ij} = \eta(\mathbf{x}_i) \cdot \eta(\mathbf{x}_j)$. Let $\mathbf{Y}_\eta \in \mathbb{R}^{N \times N}$ be the distance/similarity matrix of the $N$ data points resulted by $\eta$. Then to minimise the prediction error in Eqn. (9), $\phi$ is optimised to learn the underlying class structure embedded in $\mathbf{Y}$. As shown in the properties in Eqns. (7) and (8), $\mathbf{Y}_\eta$ can effectively preserve local proximity information when $\eta$ is set to be either the random projection-based $f$ function or the kernel method-based $g$ function. However, those proven $\eta$ is often built upon some underlying data distribution assumption, e.g., Gaussian distribution in random projection or Gaussian RBF kernel, so the $\eta$-projected features can preserve misleading proximity when the distribution assumption is inexact. In this case, $\mathbf{Y}_\eta$ is equivalent to the imperfect ground truth with partial noise. Then optimisation with Eqn. (9) is to leverage the power of neural networks to learn consistent local proximity information and rectify inconsistent proximity, resulting in a significantly optimised distance preserving space. The resulting space conveys substantially richer semantics than the $\eta$ projected space when $\mathbf{Y}_\eta$ contains sufficient genuine supervision information.

## 4 EXPERIMENTS

This section evaluates the learned representations through two typical unsupervised tasks: anomaly detection and clustering. Some preliminary results of classification can be found in Appendix H.

### 4.1 PERFORMANCE EVALUATION IN ANOMALY DETECTION

#### 4.1.1 EXPERIMENTAL SETTINGS

Our RDP model is compared with five state-of-the-art methods, including iForest (Liu et al., 2008), autoencoder (AE) (Hinton & Salakhutdinov, 2006), REPEN (Pang et al., 2018), DAGMM (Zong et al., 2018) and RND (Burda et al., 2019). iForest and AE are two of the most popular baselines. The other three methods learn representations specifically for anomaly detection.

As shown in Table 1, the comparison is performed on 14 publicly available datasets of various domains, including network intrusion, credit card fraud detection, disease detection and bank campaigning. Many of the datasets contain real anomalies, including *DDoS*, *Donors*, *Backdoor*, *Creditcard*, *Lung*, *Probe* and *U2R*. Following (Liu et al., 2008; Pang et al., 2018; Zong et al., 2018), the rare class(es) is treated as anomalies in the other datasets to create semantically real anomalies. The Area Under Receiver Operating Characteristic Curve (AUC-ROC) and the Area Under Precision-Recall Curve (AUC-PR) are used as our performance metrics. Larger AUC-ROC/AUC-PR indicates better performance. The reported performance is averaged over 10 independent runs.

Table 1: AUC-ROC (mean±std) performance of RDP and its five competing methods on 14 datasets.

| Data Characteristics | | | Our Method RDP and Its Five Competing Methods | | | | | |
|---|---|---|---|---|---|---|---|---|
| Data | N | D | Anomaly (%) | iForest | AE | REPEN | DAGMM | RND | RDP |
| **DDoS** | 464,976 | 66 | 3.75% | $0.880 \pm 0.018$ | $0.901 \pm 0.000$ | $0.933 \pm 0.002$ | $0.766 \pm 0.019$ | $0.852 \pm 0.011$ | $\mathbf{0.942 \pm 0.008}$ |
| **Donors** | 619,326 | 10 | 5.92% | $0.774 \pm 0.010$ | $0.812 \pm 0.011$ | $0.777 \pm 0.075$ | $0.763 \pm 0.110$ | $0.847 \pm 0.011$ | $\mathbf{0.962 \pm 0.011}$ |
| **Backdoor** | 95,329 | 196 | 2.44% | $0.723 \pm 0.029$ | $0.806 \pm 0.007$ | $0.857 \pm 0.001$ | $0.813 \pm 0.035$ | $\mathbf{0.935 \pm 0.002}$ | $0.910 \pm 0.021$ |
| **Ad** | 3,279 | 1,555 | 13.99% | $0.687 \pm 0.021$ | $0.703 \pm 0.000$ | $0.853 \pm 0.001$ | $0.500 \pm 0.000$ | $0.812 \pm 0.002$ | $\mathbf{0.887 \pm 0.003}$ |
| **Apascal** | 12,695 | 64 | 1.38% | $0.514 \pm 0.051$ | $0.623 \pm 0.005$ | $0.813 \pm 0.004$ | $0.710 \pm 0.020$ | $0.685 \pm 0.019$ | $\mathbf{0.823 \pm 0.007}$ |
| **Bank** | 41,188 | 62 | 11.26% | $0.713 \pm 0.021$ | $0.666 \pm 0.000$ | $0.681 \pm 0.001$ | $0.616 \pm 0.014$ | $0.690 \pm 0.006$ | $\mathbf{0.758 \pm 0.007}$ |
| **Celeba** | 202,599 | 39 | 2.24% | $0.693 \pm 0.014$ | $0.735 \pm 0.002$ | $0.802 \pm 0.002$ | $0.680 \pm 0.067$ | $0.682 \pm 0.029$ | $\mathbf{0.860 \pm 0.006}$ |
| **Census** | 299,285 | 500 | 6.20% | $0.599 \pm 0.019$ | $0.602 \pm 0.000$ | $0.542 \pm 0.003$ | $0.502 \pm 0.003$ | $\mathbf{0.661 \pm 0.003}$ | $0.653 \pm 0.004$ |
| **Creditcard** | 284,807 | 29 | 0.17% | $0.948 \pm 0.005$ | $0.948 \pm 0.000$ | $0.950 \pm 0.001$ | $0.877 \pm 0.005$ | $0.945 \pm 0.001$ | $\mathbf{0.957 \pm 0.005}$ |
| **Lung** | 145 | 3,312 | 4.13% | $0.893 \pm 0.057$ | $0.953 \pm 0.004$ | $0.949 \pm 0.002$ | $0.830 \pm 0.087$ | $0.867 \pm 0.031$ | $\mathbf{0.982 \pm 0.006}$ |
| **Probe** | 64,759 | 34 | 6.43% | $0.995 \pm 0.001$ | $0.997 \pm 0.000$ | $0.997 \pm 0.000$ | $0.953 \pm 0.008$ | $0.975 \pm 0.000$ | $\mathbf{0.997 \pm 0.000}$ |
| **R8** | 3,974 | 9,467 | 1.28% | $0.841 \pm 0.023$ | $0.835 \pm 0.000$ | $\mathbf{0.910 \pm 0.000}$ | $0.760 \pm 0.066$ | $0.883 \pm 0.006$ | $0.902 \pm 0.002$ |
| **Secom** | 1,567 | 590 | 6.63% | $0.548 \pm 0.019$ | $0.526 \pm 0.000$ | $0.510 \pm 0.004$ | $0.513 \pm 0.010$ | $0.541 \pm 0.006$ | $\mathbf{0.570 \pm 0.004}$ |
| **U2R** | 60,821 | 34 | 0.37% | $\mathbf{0.988 \pm 0.001}$ | $0.987 \pm 0.000$ | $0.978 \pm 0.000$ | $0.945 \pm 0.028$ | $0.981 \pm 0.001$ | $0.986 \pm 0.001$ |

Our RDP model uses the optional novelty loss for anomaly detection task by default. Similar to RND, given a data point $\mathbf{x}$, its anomaly score in RDP is defined as the mean squared error between the two projections resulted by $\phi(\mathbf{x}; \Theta^\star)$ and $\eta(\mathbf{x})$. Also, a boosting process is used to filter out 5% likely anomalies per iteration to iteratively improve the modelling of RDP. This is because the modelling is otherwise largely biased when anomalies are presented. In the ablation study in Section 4.1.3, we will show the contribution of all these components.

#### 4.1.2 COMPARISON TO THE STATE-OF-THE-ART COMPETING METHODS

The AUC-ROC and AUC-PR results are respectively shown in Tables 1 and 2. RDP outperforms all the five competing methods in both of AUC-ROC and AUC-PR in at least 12 out of 14 datasets. This improvement is statistically significant at the 95% confidence level according to the two-tailed sign test (Demšar, 2006). Remarkably, RDP obtains more than 10% AUC-ROC/AUC-PR improvement over the best competing method on six datasets, including *Donors*, *Ad*, *Bank*, *Celeba*, *Lung* and *U2R*. RDP can be thought as a high-level synthesis of REPEN and RND, because REPEN leverages a pairwise distance-based ranking loss to learn representations for anomaly detection while RND is built using $L_{aux}^{ad}$. In nearly all the datasets, RDP well leverages both $L_{rdp}$ and $L_{aux}^{ad}$ to achieve significant improvement over both REPEN and RND. In very limited cases, such as on datasets *Backdoor* and *Census* where RND performs very well while REPEN performs less effectively, RDP is slightly downgraded due to the use of $L_{rdp}$. In the opposite case, such as *Probe*, on which REPEN performs much better than RND, the use of $L_{aux}^{ad}$ may drag down the performance of RDP a bit.

Table 2: AUC-PR (mean±std) performance of RDP and its five competing methods on 14 datasets.

| Data | iForest | AE | REPEN | DAGMM | RND | RDP |
|---|---|---|---|---|---|---|
| **DDoS** | $0.141 \pm 0.020$ | $0.248 \pm 0.001$ | $0.300 \pm 0.012$ | $0.038 \pm 0.000$ | $0.110 \pm 0.015$ | $\mathbf{0.301 \pm 0.028}$ |
| **Donors** | $0.124 \pm 0.006$ | $0.138 \pm 0.007$ | $0.120 \pm 0.032$ | $0.070 \pm 0.024$ | $0.201 \pm 0.033$ | $\mathbf{0.432 \pm 0.061}$ |
| **Backdoor** | $0.045 \pm 0.007$ | $0.065 \pm 0.004$ | $0.129 \pm 0.001$ | $0.034 \pm 0.023$ | $\mathbf{0.433 \pm 0.015}$ | $0.305 \pm 0.008$ |
| **Ad** | $0.363 \pm 0.061$ | $0.479 \pm 0.000$ | $0.600 \pm 0.002$ | $0.140 \pm 0.000$ | $0.473 \pm 0.009$ | $\mathbf{0.726 \pm 0.007}$ |
| **Apascal** | $0.015 \pm 0.002$ | $0.023 \pm 0.001$ | $0.041 \pm 0.001$ | $0.023 \pm 0.009$ | $0.021 \pm 0.005$ | $\mathbf{0.042 \pm 0.003}$ |
| **Bank** | $0.293 \pm 0.023$ | $0.264 \pm 0.001$ | $0.276 \pm 0.001$ | $0.150 \pm 0.020$ | $0.258 \pm 0.006$ | $\mathbf{0.364 \pm 0.013}$ |
| **Celeba** | $0.060 \pm 0.006$ | $0.082 \pm 0.001$ | $0.081 \pm 0.001$ | $0.037 \pm 0.017$ | $0.068 \pm 0.010$ | $\mathbf{0.104 \pm 0.006}$ |
| **Census** | $0.071 \pm 0.004$ | $0.072 \pm 0.000$ | $0.064 \pm 0.005$ | $0.061 \pm 0.001$ | $0.081 \pm 0.001$ | $\mathbf{0.086 \pm 0.001}$ |
| **Creditcard** | $0.145 \pm 0.031$ | $\mathbf{0.382 \pm 0.004}$ | $0.359 \pm 0.014$ | $0.010 \pm 0.012$ | $0.290 \pm 0.012$ | $0.363 \pm 0.011$ |
| **Lung** | $0.379 \pm 0.092$ | $0.565 \pm 0.022$ | $0.429 \pm 0.005$ | $0.042 \pm 0.003$ | $0.381 \pm 0.104$ | $\mathbf{0.705 \pm 0.028}$ |
| **Probe** | $0.923 \pm 0.011$ | $0.964 \pm 0.002$ | $\mathbf{0.964 \pm 0.000}$ | $0.409 \pm 0.153$ | $0.609 \pm 0.014$ | $0.955 \pm 0.002$ |
| **R8** | $0.076 \pm 0.018$ | $0.097 \pm 0.006$ | $0.083 \pm 0.000$ | $0.019 \pm 0.011$ | $0.134 \pm 0.031$ | $\mathbf{0.146 \pm 0.017}$ |
| **Secom** | $0.106 \pm 0.007$ | $0.093 \pm 0.000$ | $0.091 \pm 0.001$ | $0.066 \pm 0.002$ | $0.086 \pm 0.002$ | $\mathbf{0.096 \pm 0.001}$ |
| **U2R** | $0.180 \pm 0.018$ | $0.230 \pm 0.004$ | $0.116 \pm 0.007$ | $0.025 \pm 0.019$ | $0.217 \pm 0.011$ | $\mathbf{0.261 \pm 0.005}$ |

### 4.1.3 ABLATION STUDY

This section examines the contribution of $L_{rdp}$, $L_{aux}^{ad}$ and the boosting process to the performance of RDP. The experimental results in AUC-ROC are given in Table 3, where RDP\X means the RDP variant that removes the 'X' module from RDP. In the last two columns, *Org_SS* indicates that we directly use the distance information calculated in the original space as the supervisory signal, while *SRP_SS* indicates that we use SRP to obtain the distances as the supervisory signal. It is clear that the full RDP model is the best performer. Using the $L_{rdp}$ loss only, i.e., RDP\$L_{aux}^{ad}$, can achieve performance substantially better than, or comparably well to, the five competing methods in Table 1. This is mainly because the $L_{rdp}$ loss alone can effectively force our representation learner to learn the underlying class structure on most datasets so as to minimise its prediction error. The use of $L_{aux}^{ad}$ and boosting process well complement the $L_{rdp}$ loss on the other datasets.

In terms of supervisory source, RDP and SRP_SS perform substantially better than Org_SS on most datasets. This is because the distances in both the non-linear random projection in RDP and the linear projection in SRP_SS well preserve the distance information, enabling RDP to effectively learn much more faithful class structure than that working on the original space.

Table 3: AUC-ROC results of anomaly detection (see Appendix C for similar AUC-PR results).

| Data | Decomposition | | | | Supervision Signal | |
|---|---|---|---|---|---|---|
| | RDP | RDP\$L_{rdp}$ | RDP\$L_{aux}^{ad}$ | RDP\Boosting | Org_SS | SRP_SS |
| **DDoS** | $\mathbf{0.942 \pm 0.008}$ | $0.852 \pm 0.011$ | $0.931 \pm 0.003$ | $0.866 \pm 0.011$ | $0.924 \pm 0.006$ | $0.927 \pm 0.005$ |
| **Donors** | $\mathbf{0.962 \pm 0.011}$ | $0.847 \pm 0.011$ | $0.737 \pm 0.006$ | $0.910 \pm 0.013$ | $0.728 \pm 0.005$ | $0.762 \pm 0.016$ |
| **Backdoor** | $0.910 \pm 0.021$ | $0.935 \pm 0.002$ | $0.872 \pm 0.012$ | $\mathbf{0.943 \pm 0.002}$ | $0.875 \pm 0.002$ | $0.882 \pm 0.010$ |
| **Ad** | $\mathbf{0.887 \pm 0.003}$ | $0.812 \pm 0.002$ | $0.718 \pm 0.005$ | $0.818 \pm 0.002$ | $0.696 \pm 0.003$ | $0.740 \pm 0.008$ |
| **Apascal** | $\mathbf{0.823 \pm 0.007}$ | $0.685 \pm 0.019$ | $0.732 \pm 0.007$ | $0.804 \pm 0.021$ | $0.604 \pm 0.032$ | $0.760 \pm 0.030$ |
| **Bank** | $\mathbf{0.758 \pm 0.007}$ | $0.690 \pm 0.006$ | $0.684 \pm 0.004$ | $0.736 \pm 0.009$ | $0.684 \pm 0.002$ | $0.688 \pm 0.015$ |
| **Celeba** | $\mathbf{0.860 \pm 0.006}$ | $0.682 \pm 0.029$ | $0.709 \pm 0.005$ | $0.794 \pm 0.017$ | $0.667 \pm 0.033$ | $0.734 \pm 0.027$ |
| **Census** | $0.653 \pm 0.004$ | $\mathbf{0.661 \pm 0.003}$ | $0.626 \pm 0.006$ | $0.661 \pm 0.001$ | $0.636 \pm 0.006$ | $0.560 \pm 0.006$ |
| **Creditcard** | $\mathbf{0.957 \pm 0.005}$ | $0.945 \pm 0.001$ | $0.950 \pm 0.000$ | $0.956 \pm 0.003$ | $0.947 \pm 0.001$ | $0.949 \pm 0.003$ |
| **Lung** | $\mathbf{0.982 \pm 0.006}$ | $0.867 \pm 0.031$ | $0.911 \pm 0.006$ | $0.968 \pm 0.018$ | $0.884 \pm 0.018$ | $0.928 \pm 0.008$ |
| **Probe** | $0.997 \pm 0.000$ | $0.975 \pm 0.000$ | $\mathbf{0.998 \pm 0.000}$ | $0.978 \pm 0.001$ | $0.995 \pm 0.000$ | $0.997 \pm 0.001$ |
| **R8** | $0.902 \pm 0.002$ | $0.883 \pm 0.006$ | $0.867 \pm 0.003$ | $0.895 \pm 0.004$ | $0.830 \pm 0.005$ | $\mathbf{0.904 \pm 0.005}$ |
| **Secom** | $\mathbf{0.57 \pm 0.004}$ | $0.541 \pm 0.006$ | $0.544 \pm 0.011$ | $0.563 \pm 0.006$ | $0.512 \pm 0.007$ | $0.530 \pm 0.016$ |
| **U2R** | $0.986 \pm 0.001$ | $0.981 \pm 0.001$ | $0.987 \pm 0.000$ | $\mathbf{0.988 \pm 0.002}$ | $0.987 \pm 0.000$ | $0.981 \pm 0.002$ |
| #wins/draws/losses (RDP vs.) | | 13/0/1 | 13/0/1 | 12/0/2 | 10/2/2 | 6/0/8 |

## 4.2 PERFORMANCE EVALUATION IN CLUSTERING

### 4.2.1 EXPERIMENTAL SETTINGS

For clustering, RDP is compared with four state-of-the-art unsupervised representation learning methods in four different areas, including HLLE (Donoho & Grimes, 2003) in manifold learning, Sparse Random Projection (SRP) (Li et al., 2006) in random projection, autoencoder (AE) (Hinton & Salakhutdinov, 2006) in data reconstruction-based neural network methods and Coherence Pursuit (COP) (Rahmani & Atia, 2017) in robust PCA. These representation learning methods are first used to yield the new representations, and K-means (Hartigan & Wong, 1979) is then applied to the

representations to perform clustering. Two widely-used clustering performance metrics, Normalised Mutual Info (NMI) score and F-score, are used. Larger NMI or F-score indicates better performance. The clustering performance in the original feature space, denoted as Org, is used as a baseline. As shown in Table 4, five high-dimensional real-world datasets are used. Some of the datasets are image/text data. Since here we focus on the performance on tabular data, they are converted into tabular data using simple methods, i.e., by treating each pixel as a feature unit for image data or using bag-of-words representation for text data[2]. The reported NMI score and F-score are averaged over 30 times to address the randomisation issue in K-means clustering. In this section RDP adds the reconstruction loss $L_{aux}^{clu}$ by default, but RDP also works very well without the use of $L_{aux}^{clu}$.

### 4.2.2 COMPARISON TO THE-STATE-OF-THE-ART COMPETING METHODS

Table 4 shows the NMI and F-score performance of K-means clustering. Our method RDP enables K-means to achieve the best performance on three datasets and ranks second in the other two datasets. RDP-enabled clustering performs substantially and consistently better than that based on AE in terms of both NMI and F-score. This demonstrates that the random distance loss enables RDP to effectively capture some class structure in the data which cannot be captured by using the reconstruction loss. RDP also consistently outperforms the random projection method, SRP, and the robust PCA method, COP. It is interesting that K-means clustering performs best in the original space on *Sector*. This may be due to that this data contains many relevant features, resulting in no obvious curse of dimensionality issue. *Olivetti* may contain complex manifolds which require extensive neighbourhood information to find them, so only HLLE can achieve this goal in such cases. Nevertheless, RDP performs much more stably than HLLE across the five datasets.

Table 4: NMI and F-score performance of K-means on the original space and projected spaces.

| Data Characteristics | | | NMI Performance | | | | | |
|---|---|---|---|---|---|---|---|---|
| Data | N | D | Org | HLLE | SRP | AE | COP | RDP |
| R8 | 7,674 | 17,387 | $0.524 \pm 0.047$ | $0.004 \pm 0.001$ | $0.459 \pm 0.031$ | $0.471 \pm 0.043$ | $0.025 \pm 0.003$ | $\mathbf{0.539 \pm 0.040}$ |
| 20news | 18,846 | 130,107 | $0.080 \pm 0.004$ | $0.017 \pm 0.000$ | $0.075 \pm 0.002$ | $0.075 \pm 0.006$ | $0.027 \pm 0.040$ | $\mathbf{0.084 \pm 0.005}$ |
| Olivetti | 400 | 4,096 | $0.778 \pm 0.014$ | $\mathbf{0.841 \pm 0.011}$ | $0.774 \pm 0.011$ | $0.782 \pm 0.010$ | $0.333 \pm 0.018$ | $0.805 \pm 0.012$ |
| Sector | 9,619 | 55,197 | $\mathbf{0.336 \pm 0.008}$ | $0.122 \pm 0.004$ | $0.273 \pm 0.011$ | $0.253 \pm 0.010$ | $0.129 \pm 0.014$ | $0.305 \pm 0.007$ |
| RCV1 | 20,242 | 47,236 | $0.154 \pm 0.000$ | $0.006 \pm 0.000$ | $0.134 \pm 0.024$ | $0.146 \pm 0.010$ | N/A | $\mathbf{0.165 \pm 0.000}$ |
| Data Characteristics | | | F-score Performance | | | | | |
| Data | N | D | Org | HLLE | SRP | AE | COP | RDP |
| R8 | 7,674 | 17,387 | $0.185 \pm 0.189$ | $0.085 \pm 0.000$ | $0.317 \pm 0.045$ | $0.312 \pm 0.068$ | $0.088 \pm 0.002$ | $\mathbf{0.360 \pm 0.055}$ |
| 20news | 18,846 | 130,107 | $0.116 \pm 0.006$ | $0.007 \pm 0.000$ | $0.109 \pm 0.006$ | $0.083 \pm 0.010$ | $0.009 \pm 0.004$ | $\mathbf{0.119 \pm 0.006}$ |
| Olivetti | 400 | 4,096 | $0.590 \pm 0.029$ | $\mathbf{0.684 \pm 0.024}$ | $0.579 \pm 0.022$ | $0.602 \pm 0.023$ | $0.117 \pm 0.011$ | $0.638 \pm 0.026$ |
| Sector | 9,619 | 55,197 | $\mathbf{0.208 \pm 0.008}$ | $0.062 \pm 0.001$ | $0.187 \pm 0.009$ | $0.184 \pm 0.010$ | $0.041 \pm 0.004$ | $0.191 \pm 0.007$ |
| RCV1 | 20,242 | 47,236 | $0.519 \pm 0.000$ | $0.342 \pm 0.000$ | $0.508 \pm 0.003$ | $0.514 \pm 0.057$ | N/A | $\mathbf{0.572 \pm 0.003}$ |

Table 5: F-score performance of K-means clustering (see similar NMI results in Appendix D).

| | Decomposition | | | Supervision Signal | |
|---|---|---|---|---|---|
| Data | RDP | RDP$\backslash L_{rdp}$ | RDP$\backslash L_{aux}^{clu}$ | Org_SS | SRP_SS |
| R8 | $0.360 \pm 0.055$ | $0.312 \pm 0.068$ | $0.330 \pm 0.052$ | $0.359 \pm 0.028$ | $\mathbf{0.363 \pm 0.046}$ |
| 20news | $\mathbf{0.119 \pm 0.006}$ | $0.083 \pm 0.010$ | $0.117 \pm 0.007$ | $0.111 \pm 0.005$ | $0.111 \pm 0.007$ |
| Olivetti | $\mathbf{0.638 \pm 0.026}$ | $0.602 \pm 0.023$ | $0.597 \pm 0.019$ | $0.610 \pm 0.022$ | $0.601 \pm 0.023$ |
| Sector | $0.191 \pm 0.007$ | $0.184 \pm 0.010$ | $\mathbf{0.217 \pm 0.007}$ | $0.181 \pm 0.007$ | $0.186 \pm 0.009$ |
| RCV1 | $\mathbf{0.572 \pm 0.003}$ | $0.514 \pm 0.057$ | $0.526 \pm 0.011$ | $0.523 \pm 0.003$ | $0.532 \pm 0.001$ |

### 4.2.3 ABLATION STUDY

Similar to anomaly detection, this section examines the contribution of the two loss functions $L_{rdp}$ and $L_{aux}^{clu}$ to the performance of RDP, as well as the impact of different supervisory sources on the performance. The F-score results of this experiment are shown in Table 5, in which the notations have exactly the same meaning as in Table 3. The full RDP model that uses both $L_{rdp}$ and $L_{aux}^{clu}$ performs more favourably than its two variants, RDP$\backslash L_{rdp}$ and RDP$\backslash L_{aux}^{clu}$, but it is clear that using $L_{rdp}$ only performs very comparably to the full RDP. However, using $L_{aux}^{clu}$ only may result in large

---

[2]RDP can also build upon advanced representation learning methods for the data transformation, for which some interesting preliminary results are presented in Appendix G.

performance drops in some datasets, such as *R8*, *20news* and *Olivetti*. This indicates $L_{rdp}$ is a more important loss function to the overall performance of the full RDP model. In terms of supervisory source, distances obtained by the non-linear random projection in RDP are much more effective than the two other sources on some datasets such as *Olivetti* and *RCV1*. Three different supervisory sources are very comparable on the other three datasets.

## 5 RELATED WORK

**Self-supervised Learning.** Self-supervised learning has been recently emerging as one of the most popular and effective approaches for representation learning. Many of the self-supervised methods learn high-level representations by predicting some sort of 'context' information, such as spatial or temporal neighbourhood information. For example, the popular distributed representation learning techniques in NLP, such as CBOW/skip-gram (Mikolov et al., 2013a) and phrase/sentence embeddings in (Mikolov et al., 2013b; Le & Mikolov, 2014; Hill et al., 2016), learn the representations by predicting the text pieces (e.g., words/phrases/sentences) using its surrounding pieces as the context. In image processing, the pretext task can be the prediction of a patch of missing pixels (Pathak et al., 2016; Zhang et al., 2017) or the relative position of two patches (Doersch et al., 2015). Also, a number of studies (Goroshin et al., 2015; Misra et al., 2016; Lee et al., 2017; Oord et al., 2018) explore temporal contexts to learn representations from video data, e.g., by learning the temporal order of sequential frames. Some other methods (Agrawal et al., 2015; Zhou et al., 2017; Gidaris et al., 2018) are built upon a discriminative framework which aims at discriminating the images before and after some transformation, e.g., ego motion in video data (Agrawal et al., 2015; Zhou et al., 2017) and rotation of images (Gidaris et al., 2018). There have also been popular to use generative adversarial networks (GANs) to learn features (Radford et al., 2015; Chen et al., 2016). The above methods have demonstrated powerful capability to learn semantic representations. However, most of them use the supervisory signals available in image/video data only, which limits their application into other types of data, such as traditional tabular data. Although our method may also work on image/video data, we focus on handling high-dimensional tabular data to bridge this gap.

**Other Approaches.** There have been several well-established unsupervised representation learning approaches for handling tabular data, such as random projection (Arriaga & Vempala, 1999; Bingham & Mannila, 2001; Li et al., 2006), PCA (Wold et al., 1987; Schölkopf et al., 1997; Rahmani & Atia, 2017), manifold learning (Roweis & Saul, 2000; Donoho & Grimes, 2003; Hinton & Roweis, 2003; McInnes et al., 2018) and autoencoder (Hinton & Salakhutdinov, 2006; Vincent et al., 2010). One notorious issue of PCA or manifold learning approaches is their prohibitive computational cost in dealing with large-scale high-dimensional data due to the costly neighbourhood search and/or eigen decomposition. Random projection is a computationally efficient approach, supported by proven distance preservation theories such as the Johnson-Lindenstrauss lemma (Johnson & Lindenstrauss, 1984). We show that the preserved distances by random projection can be harvested to effectively supervise the representation learning. Autoencoder networks are another widely-used efficient feature learning approach which learns low-dimensional representations by minimising reconstruction errors. One main issue with autoencoders is that they focus on preserving global information only, which may result in loss of local structure information. Some representation learning methods are specifically designed for anomaly detection (Pang et al., 2018; Zong et al., 2018; Burda et al., 2019). By contrast, we aim at generic representations learning while being flexible to incorporate optionally task-dependent losses to learn task-specific semantic-rich representations.

## 6 CONCLUSION

We introduce a novel Random Distance Prediction (RDP) model which learns features in a fully unsupervised fashion by predicting data distances in a randomly projected space. The key insight is that random mapping is a theoretical proven approach to obtain approximately preserved distances, and to well predict these random distances, the representation learner is optimised to learn consistent preserved proximity information while at the same time rectifying inconsistent proximity, resulting in representations with optimised distance preserving. Our idea is justified by thorough experiments in two unsupervised tasks, anomaly detection and clustering, which show RDP-enabled anomaly detectors and clustering substantially outperform their counterparts on 19 real-world datasets. We plan to extend RDP to other types of data to broaden its application scenarios.

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

## A    IMPLEMENTATION DETAILS

**RDP-enabled Anomaly Detection.** The RDP consists of one fully connected layer with 50 hidden units, followed by a leaky-ReLU layer. It is trained using Stochastic Gradient Descent (SGD) as its optimiser for 200 epochs, with 192 samples per batch. The learning rate is fixed to 0.1. We repeated the boosting process 30 times to obtain statistically stable results. In order to have fair comparisons, we also adapt the competing methods AE, REPEN, DAGMM and RND into ensemble methods and perform the experiments using an ensemble size of 30.

**RDP-enabled Clustering.** RDP uses a similar network architecture and optimisation settings as the one used in anomaly detection, i.e., the network consists of one fully connected layer, followed by a leaky-ReLU layer, which is optimised by SGD with 192 samples per batch and 0.1 learning rate. Compared to anomaly detection, more semantic information is required for clustering algorithms to work well, so the network consists of 1,024 hidden units and is trained for 1,000 epochs. Clustering is a significant yet common analysis method, which aims at grouping samples close to each other into the same clusters and separating far away data points into different clusters. Compared to anomaly detection that often requires pattern frequency information, clustering has a higher requirement of the representation expressiveness. Therefore, if the representative ability of a model is strong enough, it should also be able to learn representations that enable clustering to work well on the projected space.

Note that the representation dimension $M$ in the $\phi$ function and the projection dimension $K$ in the $\eta$ function are set to be the same to alleviate parameter tuning. This means that $M = K = 50$ is used in anomaly detection and $M = K = 1024$ is used in clustering. We have also tried deeper network structures, but they worked less effectively than the shallow networks in both anomaly detection and clustering. This may be because the supervisory signal is not strong enough to effectively learn deeper representations. We show in Appendix E that RDP performs stably w.r.t. a range of representation dimensions in both anomaly detection and clustering tasks.

The runtime of RDP at the testing stage is provided in Appendix F with that of the competing methods as baselines. For both anomaly detection and clustering tasks, RDP achieves very comparable time complexity to the most efficient competing methods (see Tables 10 and 11 in Appendix F for detail).

## B    DATASETS

The statistics and the accessible links of the datasets used in the anomaly detection and clustering tasks are respectively presented in Tables 6 and 7. *DDoS* is a dataset containing DDoS attacks and normal network flows. *Donors* is from KDD Cup 2014, which is used for detecting a very small number of outstanding donors projects. *Backdoor* contains backdoor network attacks derived from the UNSW-NB15 dataset. *Creditcard* is a credit card fraud detection dataset. *Lung* contains data records of lung cancer patients and normal patients. *Probe* and *U2R* are derived from KDD Cup 99, in which probing and user-to-root attacks are respectively used as anomalies against the normal network flows. The above datasets contain real anomalies. Following (Liu et al., 2008; Pang et al., 2018; Zong et al., 2018), the other anomaly detection datasets are transformed from classification datasets by using the rare class(es) as the anomaly class, which generates semantically real anomalies.

Table 6: Datasets used in the anomaly detection task

| Data | N | D | Anomaly (%) | Link |
|---|---|---|---|---|
| DDoS | 464,976 | 66 | 3.75% | http://www.csmining.org/cdmc2018/index.php |
| Donors | 619,326 | 10 | 5.92% | https://www.kaggle.com/c/kdd-cup-2014-predicting-excitement-at-donors-choose |
| Backdoor | 95,329 | 196 | 2.44% | https://www.unsw.adfa.edu.au/unsw-canberra-cyber/cybersecurity |
| Ad | 3,279 | 1,555 | 13.99% | https://archive.ics.uci.edu/ml/datasets/internet+advertisements |
| Apascal | 12,695 | 64 | 1.38% | http://vision.cs.uiuc.edu/attributes/ |
| Bank | 41,188 | 62 | 11.26% | https://archive.ics.uci.edu/ml/datasets/Bank+Marketing |
| Celeba | 202,599 | 39 | 2.24% | http://mmlab.ie.cuhk.edu.hk/projects/CelebA.html |
| Census | 299,285 | 500 | 6.20% | https://archive.ics.uci.edu/ml/datasets/Census-Income+%28KDD%29 |
| Creditcard | 284,807 | 29 | 0.17% | https://www.kaggle.com/mlg-ulb/creditcardfraud |
| Lung | 145 | 3,312 | 4.13% | https://archive.ics.uci.edu/ml/datasets/Lung+Cancer |
| Probe | 64,759 | 34 | 6.43% | http://kdd.ics.uci.edu/databases/kddcup99/kddcup99.html |
| R8 | 3,974 | 9,467 | 1.28% | http://csmining.org/tl_files/Project_Datasets/r8_r52/r8-train-all-terms.txt |
| Secom | 1,567 | 590 | 6.63% | https://archive.ics.uci.edu/ml/datasets/secom |
| U2R | 60,821 | 34 | 0.37% | http://kdd.ics.uci.edu/databases/kddcup99/kddcup99.html |

*R8*, *20news*, *Sector* and *RCV1* are widely used text classification benchmark datasets. *Olivetti* is a widely-used face recognition dataset.

Table 7: Datasets used in the clustering task

| Data | N | D | #Classes | Link |
|---|---|---|---|---|
| R8 | 7,674 | 17,387 | 8 | http://csmining.org/tl_files/Project_Datasets/r8_r52/r8-train-all-terms.txt |
| 20news | 18,846 | 130,107 | 20 | https://scikit-learn.org/0.19/datasets/twenty_newsgroups.html |
| Olivetti | 400 | 4,096 | 40 | https://scikit-learn.org/0.19/datasets/olivetti_faces.html |
| Sector | 9,619 | 55,197 | 105 | https://www.csie.ntu.edu.tw/~cjlin/libsvmtools/datasets/multiclass.html#sector |
| RCV1 | 20,242 | 47,236 | 2 | https://www.csie.ntu.edu.tw/~cjlin/libsvmtools/datasets/binary.html#rcv1.binary |

## C  AUC-PR Performance of Ablation Study in Anomaly detection

The experimental results of AUC-PR performance of RDP and its variants in the anomaly detection task are shown in Table 8. Similar to the results shown in Table 3, using the $L_{rdp}$ loss only, our proposed RDP model can achieve substantially better performance over its counterparts. By removing the $L_{rdp}$ loss, the performance of RDP drops significantly in 11 out of 14 datasets. This demonstrates that the $L_{rdp}$ loss is heavily harvested by our RDP model to learn high-quality representations from random distances. Removing $L_{aux}^{ad}$ from RDP also results in substantial loss of AUC-PR in many datasets. This indicates both the random distance prediction loss $L_{rdp}$ and the task-dependent loss $L_{aux}^{ad}$ are critical to RDP. The boosting process is also important, but is not as critical as the two losses. Consistent with the observations derived from Table 3, distances calculated in non-linear and linear random mapping spaces are more effective supervisory sources than that in the original space.

## D  NMI Performance of Ablation Study in Clustering

Table 9 shows the NMI performance of RDP and its variants in the clustering task. It is clear that our RDP model with the $L_{rdp}$ loss is able to achieve NMI performance that is comparably well to the full RDP model, which is consistent to the observations in Table 5. Without using the $L_{rdp}$ loss, the performance of the RDP model has some large drops on nearly all the datasets. This reinforces the crucial importance of $L_{rdp}$ to RDP, which also justifies that using $L_{rdp}$ alone RDP can learn expressive representations. Similar to the results in Table 5, RDP is generally more reliable supervisory sources than Org_SS and SRP_SS in this set of results.

## E  Sensitivity w.r.t. the Dimensionality of Representation Space

This section presents the performance of RDP using different representation dimensions in its feature learning layer. The sensitivity test is performed for both anomaly detection and clustering tasks.

Table 8: AUC-PR performance of RDP and its variants in the anomaly detection task.

| Data | RDP | Decomposition | | | Supervision Signal | |
|---|---|---|---|---|---|---|
| | RDP | RDP$\backslash L_{rdp}$ | RDP$\backslash L_{aux}^{ad}$ | RDP$\backslash$Boosting | Org_SS | SRP_SS |
| **DDoS** | $0.301 \pm 0.028$ | $0.110 \pm 0.015$ | $0.364 \pm 0.013$ | $0.114 \pm 0.001$ | $0.363 \pm 0.007$ | $\mathbf{0.380 \pm 0.030}$ |
| **Donors** | $\mathbf{0.432 \pm 0.061}$ | $0.201 \pm 0.033$ | $0.104 \pm 0.007$ | $0.278 \pm 0.040$ | $0.099 \pm 0.004$ | $0.113 \pm 0.010$ |
| **Backdoor** | $0.305 \pm 0.008$ | $0.433 \pm 0.015$ | $0.142 \pm 0.006$ | $\mathbf{0.537 \pm 0.005}$ | $0.143 \pm 0.005$ | $0.154 \pm 0.028$ |
| **Ad** | $\mathbf{0.726 \pm 0.007}$ | $0.473 \pm 0.009$ | $0.491 \pm 0.014$ | $0.488 \pm 0.008$ | $0.419 \pm 0.015$ | $0.530 \pm 0.007$ |
| **Apascal** | $\mathbf{0.042 \pm 0.003}$ | $0.021 \pm 0.005$ | $0.031 \pm 0.002$ | $0.028 \pm 0.003$ | $0.016 \pm 0.003$ | $0.035 \pm 0.007$ |
| **Bank** | $\mathbf{0.364 \pm 0.013}$ | $0.258 \pm 0.006$ | $0.266 \pm 0.018$ | $0.278 \pm 0.007$ | $0.262 \pm 0.016$ | $0.265 \pm 0.021$ |
| **Celeba** | $\mathbf{0.104 \pm 0.006}$ | $0.068 \pm 0.010$ | $0.060 \pm 0.004$ | $0.072 \pm 0.008$ | $0.050 \pm 0.009$ | $0.065 \pm 0.010$ |
| **Census** | $0.086 \pm 0.001$ | $0.081 \pm 0.001$ | $0.075 \pm 0.001$ | $\mathbf{0.087 \pm 0.001}$ | $0.077 \pm 0.002$ | $0.064 \pm 0.001$ |
| **Creditcard** | $0.363 \pm 0.011$ | $0.290 \pm 0.012$ | $\mathbf{0.414 \pm 0.02}$ | $0.329 \pm 0.007$ | $0.362 \pm 0.016$ | $0.372 \pm 0.024$ |
| **Lung** | $\mathbf{0.705 \pm 0.028}$ | $0.381 \pm 0.104$ | $0.437 \pm 0.083$ | $0.542 \pm 0.139$ | $0.361 \pm 0.054$ | $0.464 \pm 0.053$ |
| **Probe** | $0.955 \pm 0.002$ | $0.609 \pm 0.014$ | $0.952 \pm 0.007$ | $0.628 \pm 0.011$ | $0.937 \pm 0.005$ | $\mathbf{0.959 \pm 0.011}$ |
| **R8** | $0.146 \pm 0.017$ | $0.134 \pm 0.031$ | $0.109 \pm 0.006$ | $\mathbf{0.173 \pm 0.028}$ | $0.067 \pm 0.016$ | $0.134 \pm 0.019$ |
| **Secom** | $\mathbf{0.096 \pm 0.001}$ | $0.086 \pm 0.002$ | $0.096 \pm 0.006$ | $0.090 \pm 0.001$ | $0.088 \pm 0.004$ | $0.093 \pm 0.004$ |
| **U2R** | $0.261 \pm 0.005$ | $0.217 \pm 0.011$ | $\mathbf{0.266 \pm 0.007}$ | $0.238 \pm 0.009$ | $0.187 \pm 0.013$ | $0.239 \pm 0.023$ |
| #wins/draws/losses (RDP vs.) | | 13/0/1 | 11/0/3 | 11/0/3 | 12/0/2 | 5/0/9 |

Table 9: NMI performance of RDP and its variants in the clustering task.

| Data | RDP | Decomposition | | Supervision Signal | |
|---|---|---|---|---|---|
| | RDP | RDP$\backslash L_{rdp}$ | RDP$\backslash L_{aux}^{clu}$ | Org_SS | SRP_SS |
| **R8** | $0.539 \pm 0.040$ | $0.471 \pm 0.043$ | $0.505 \pm 0.037$ | $0.567 \pm 0.021$ | $\mathbf{0.589 \pm 0.039}$ |
| **20news** | $\mathbf{0.084 \pm 0.005}$ | $0.075 \pm 0.006$ | $0.081 \pm 0.002$ | $0.075 \pm 0.002$ | $0.074 \pm 0.003$ |
| **Olivetti** | $\mathbf{0.805 \pm 0.012}$ | $0.782 \pm 0.010$ | $0.784 \pm 0.010$ | $0.795 \pm 0.011$ | $0.787 \pm 0.011$ |
| **Sector** | $0.305 \pm 0.007$ | $0.253 \pm 0.010$ | $\mathbf{0.340 \pm 0.007}$ | $0.295 \pm 0.009$ | $0.298 \pm 0.008$ |
| **Rcv1** | $0.165 \pm 0.000$ | $0.146 \pm 0.010$ | $\mathbf{0.168 \pm 0.000}$ | $0.154 \pm 0.002$ | $0.147 \pm 0.000$ |

## E.1 SENSITIVITY TEST IN ANOMALY DETECTION

Figures 2 and 3 respectively show the AUC-ROC and AUC-PR performance of RDP using different representation dimensions on all the 14 anomaly detection datasets used in this work. It is clear from both performance measures that RDP generally performs stably w.r.t. the use of different representation dimensions on diverse datasets. This demonstrates the general stability of our RDP method on different application domains. On the other hand, the flat trends also indicate that, as an unsupervised learning source, the random distance cannot provide sufficient supervision information to learn richer and more complex representations in a higher-dimensional space. This also explains the performance on quite a few datasets where the performance of RDP decreases when increasing the representation dimension. In general, the representation dimension 50 is recommended for RDP to achieve effective anomaly detection on datasets from different domains.

## E.2 SENSITIVITY TEST IN CLUSTERING

Figure 4 presents the NMI and F-score performance of RDP-enabled K-means clustering using different representation dimensions on all the five datasets in the clustering task. Similar to the sensitivity test results in the anomaly detection task, on all the five datasets, K-means clustering performs stably in the representation space resulted by RDP with different representation dimensions. The clustering performance may drop a bit when the representation dimension is relatively low, e.g., 512. Increasing the representation to 1,280 may help RDP gain better representation power in some datasets but is not a consistently better choice. Thus, the representation dimension 1,024 is generally recommended for clustering. Recall that the required representation dimension in clustering is normally significantly higher than that in anomaly detection, because clustering generally requires significantly more information to perform well than anomaly detection.

## F COMPUTATIONAL EFFICIENCY

The runtime of RDP is compared with its competing methods in both anomaly detection and clustering tasks. Since training time can vary significantly using different training strategies in deep learning-based methods, it is difficult to have a fair comparison of the training time. Moreover, the

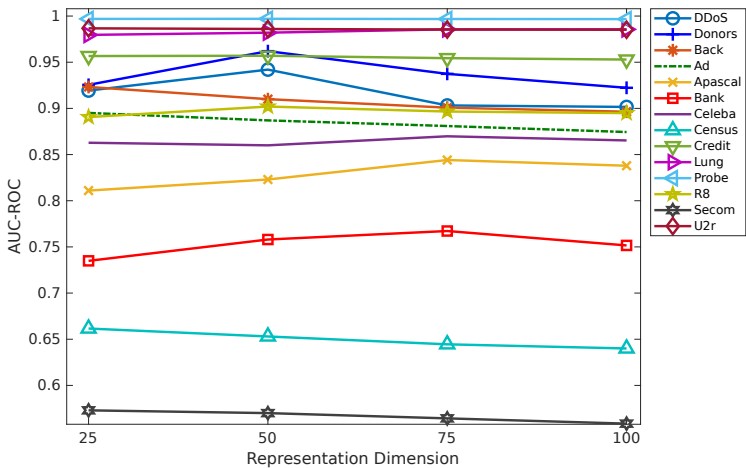

Figure 2: AUC-ROC results of RDP w.r.t. different representation dimensions on 14 datasets.

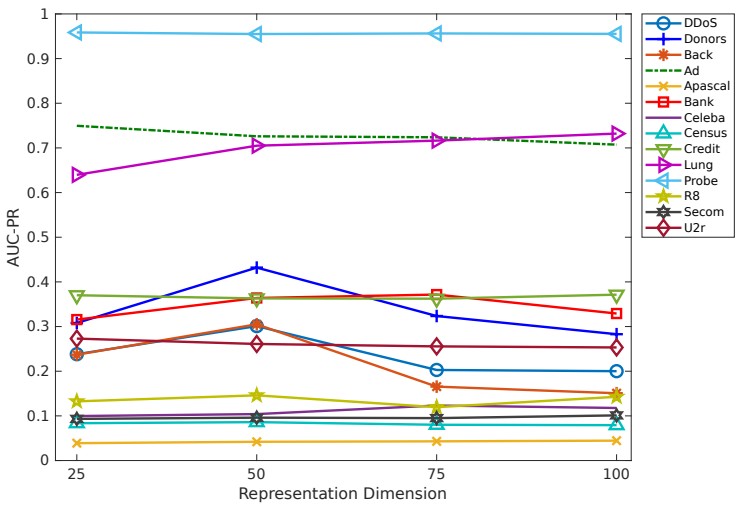

Figure 3: AUC-PR results of RDP w.r.t. different representation dimensions on 14 datasets.

models can often be trained offline. Thus, we focus on comparing the runtime at the testing stage. All the runtime experiments below were done on a computing server node equipped with 32 Intel Xeon E5-2680 CPUs (2.70GHz) and 128GB Random Access Memory.

### F.1 TESTING RUNTIME IN ANOMALY DETECTION

The testing runtime in seconds of RDP and its five competing anomaly detection methods on 14 anomaly detection datasets are provided in Table 10. Since most of the methods integrate representation learning and anomaly detection into a single framework, the runtime includes the execution time of feature learning and anomaly detection for all six methods. In general, on most large datasets, RDP runs comparably fast to the most efficient methods iForest and RND, and is faster the two recently proposed deep methods REPEN and DAGMM. Particularly, RDP runs faster than REPEN and DAGMM by a factor of around five on high-dimensional and large-scale datasets like Donors and Census. RDP is slower than the competing methods in processing small datasets. This is mainly

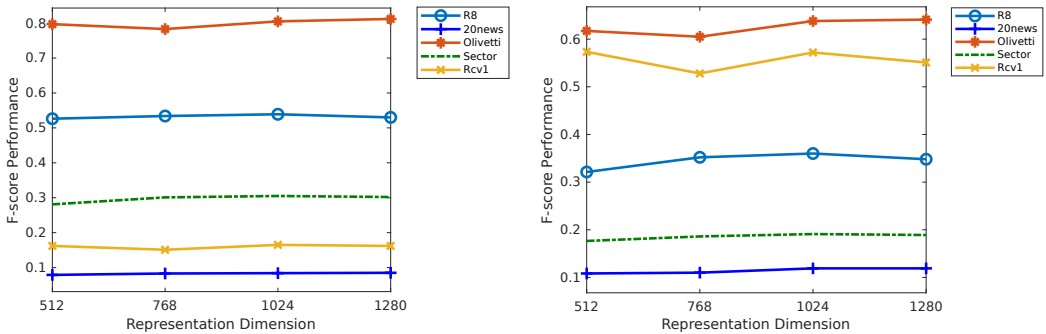

Figure 4: NMI and F-score performance of RDP-enabled K-means using different representation dimensions on all the five datasets used in clustering.

because RDP has a base runtime of its boosting process. Therefore, the runtime of RDP seems to be almost constant across the datasets. This is a very desired property for handling high-dimensional and large-scale datasets.

Table 10: Testing runtime (in seconds) on 14 anomaly detection datasets.

| Data Characteristics | | | RDP and Its Five Competing Methods | | | | | |
|---|---|---|---|---|---|---|---|---|
| Data | N | D | iForest | AE | REPEN | DAGMM | RND | RDP |
| DDoS | 464,976 | 66 | 54.06 | 86.86 | 172.47 | 197.85 | 31.86 | 28.93 |
| Donors | 619,326 | 10 | 28.17 | 52.31 | 226.14 | 194.45 | 44.31 | 36.84 |
| Backdoor | 95,329 | 196 | 26.51 | 51.66 | 36.43 | 187.61 | 12.26 | 29.95 |
| Ad | 3,279 | 1,555 | 6.71 | 14.71 | 3.24 | 31.54 | 8.12 | 30.83 |
| Apascal | 12,695 | 64 | 6.53 | 4.27 | 6.30 | 69.35 | 3.62 | 22.88 |
| Bank | 41,188 | 62 | 9.72 | 6.87 | 17.25 | 170.56 | 9.31 | 28.47 |
| Celeba | 202,599 | 39 | 20.54 | 26.70 | 71.60 | 223.70 | 18.05 | 33.91 |
| Census | 299,285 | 500 | 155.77 | 225.29 | 121.08 | 236.21 | 42.83 | 57.74 |
| Creditcard | 284,807 | 29 | 22.45 | 29.38 | 103.18 | 235.93 | 20.97 | 30.84 |
| Lung | 145 | 3,312 | 6.20 | 13.11 | 2.16 | 39.75 | 1.44 | 24.29 |
| Probe | 64,759 | 34 | 9.55 | 10.06 | 28.14 | 131.40 | 9.90 | 29.61 |
| R8 | 3,974 | 9,467 | 59.70 | 45.48 | 7.81 | 31.99 | 8.26 | 14.33 |
| Secom | 1,567 | 590 | 7.32 | 5.78 | 2.83 | 18.22 | 3.23 | 22.52 |
| U2R | 60,821 | 34 | 8.95 | 9.38 | 26.55 | 185.88 | 9.90 | 28.10 |

## F.2 TESTING RUNTIME IN CLUSTERING

Table 11 shows the testing runtime of RDP and its four competing methods in enabling clustering on five datasets. Since exactly the same K-means clustering is used on the features in all the five cases, we exclude the runtime of the K-means clustering for more straightforward comparison. The results show that RDP runs comparably fast to the very efficient methods SRP and AE since they do not involve complex computation at the testing stage; RDP runs about five orders of magnitude faster than HLLE since HLLE takes a huge amount of time in its nearest neighbours searching. Note that 'Org' indicates the clustering performed on the original space, so it involves no feature learning and does not take any time.

Table 11: Testing runtime (in seconds) on five clustering datasets.

| Data Characteristics | | | RDP and Its Four Competing Methods | | | | |
|---|---|---|---|---|---|---|---|
| Data | N | D | Org | HLLE | SRP | AE | RDP |
| R8 | 7,674 | 17,387 | - | 9,658.85 | 1.16 | 1.08 | 0.89 |
| 20news | 18,846 | 130,107 | - | 94,349.20 | 2.26 | 11.49 | 6.85 |
| Olivetti | 400 | 4,096 | - | 166.02 | 0.73 | 0.03 | 0.03 |
| Sector | 9,619 | 55,197 | - | 24,477.80 | 1.40 | 4.28 | 2.87 |
| RCV1 | 20,242 | 47,236 | - | 47,584.79 | 2.80 | 8.91 | 5.04 |

# G    COMPARISON TO STATE-OF-THE-ART REPRESENTATION LEARNING METHODS FOR RAW TEXT AND IMAGE DATA

Since RDP relies on distance information as its supervisory signal, one interesting question is that, can RDP still work when the presented data is raw data in a non-Euclidean space, such as raw text and image data? One simple and straightforward way to enable RDP to handle those raw data is, as what we did on the text and image data used in the evaluation of clustering, to first convert the raw texts/images into feature vectors using commonly-used methods, e.g., TF-IDF (Aizawa, 2003) for text data and treating each pixel as a feature unit for image data, and then perform RDP on these vector spaces. A further question is that, do we need RDP in handling those data since there are now a large number of advanced representation learning methods that are specifically designed for raw text/image datasets? Or, how is the performance of RDP compared to those advanced representation learning methods for raw text/image datasets? This section provides some preliminary results in the clustering task for answering these questions.

## G.1    ON RAW TEXT DATA

On the raw text datasets R8 and 20news, we first compare RDP with the advanced document representation method Doc2Vec[3] as in (Le & Mikolov, 2014). Recall that, for RDP, we first use the bag-of-words model and document frequency information (e.g., TF-IDF) to simply convert documents into high-dimensional feature vectors and then perform RDP using the feature vectors. Doc2Vec leverages the idea of distributed representations to directly learn representations of documents. We further derive a variant of RDP, namely Doc2Vec+RDP, which performs RDP on the Doc2Vec projected representation space rather than the bag-of-words vector space. All RDP, Doc2Vec and Doc2Vec+RDP project data onto a 1,024-dimensional space for the subsequent learning tasks. Note that, for the method Doc2Vec+RDP, to better examine the capability of RDP in exploiting the Doc2Vec projected space, we first use Doc2Vec project raw text data onto a higher-dimensional space (5,120 dimensions for R8 and 10,240 dimensions for 20news), and RDP further learns a 1,024-dimensional space from this higher-dimensional space.

The comparison results are shown in Table 12. Two interesting observations can be seen. First, RDP can significantly outperform Doc2Vec on *R8* or performs comparably well on *20news*. This may be due to the fact that the local proximity information learned in RDP is critical to clustering; although the word prediction approach in Doc2Vec helps learn semantic-rich representations for words/sentences/paragraphs, the pairwise document distances may be less effective than RDP since Doc2Vec is not like RDP that is designed to optimise this proximity information. Second, Doc2Vec+RDP can achieve substantially better performance than Doc2Vec, especially on the dataset *20news* where Doc2Vec+RDP achieves a NMI score of 0.198 while that of Doc2Vec is only 0.084. This may be because, as discussed in Section 3.3, RDP is equivalent to learn an optimised feature space out of its input space (Doc2Vec projected feature space in this case) using imperfect supervision information. When there is sufficient accurate supervision information, RDP can learn a substantially better feature space than its input space. This is also consistent with the results in Table 4, in which clustering based on the RDP projected space also performs substantially better than that working in the original space 'Org'.

Table 12: NMI and F-score performance of K-means clustering using RDP, Doc2Vec, and Doc2Vec+RDP based feature representations of the text datasets R8 and news20.

| Data Characteristics | | | NMI Performance | | |
|---|---|---|---|---|---|
| **Data** | **N** | **D** | **Doc2Vec** | **RDP** | **Doc2Vec+RDP** |
| **R8** | 7,674 | 17,387 | $0.241 \pm 0.022$ | $\mathbf{0.539 \pm 0.040}$ | $0.250 \pm 0.003$ |
| **20news** | 18,846 | 130,107 | $0.080 \pm 0.003$ | $0.084 \pm 0.005$ | $\mathbf{0.198 \pm 0.009}$ |
| Data Characteristics | | | F-score Performance | | |
| **Data** | **N** | **D** | **Doc2Vec** | **RDP** | **Doc2Vec+RDP** |
| **R8** | 7,674 | 17,387 | $0.317 \pm 0.014$ | $\mathbf{0.360 \pm 0.055}$ | $0.316 \pm 0.007$ |
| **20news** | 18,846 | 130,107 | $0.115 \pm 0.006$ | $0.119 \pm 0.006$ | $\mathbf{0.126 \pm 0.009}$ |

---

[3]We use the implementation of Doc2Vec in a popular text mining python package `gensim` available at https://radimrehurek.com/gensim/index.html

## G.2   On Raw Image Data

On the raw image dataset *Olivetti*, we compare RDP with the advanced representation learning method for raw images, RotNet (Gidaris et al., 2018). RDP uses each image pixel as a feature unit and performs on a $64 \times 64$ vector space. RotNet directly learns representations of images by predicting whether a given image is rotated or not. Similar to the experiments on raw text data, we also evaluate the performance of RDP working on the RotNet projected space, i.e., RotNet+RDP. All RDP, RotNet and RotNet+RDP first learn a 1,024 representation space, and then K-means is applied to the learned space to perform clustering. In the case of RotNet+RDP, the raw image data is first projected onto a 2,048-dimensional space, and then RDP is applied to this higher-dimensional space to learn a 1,024-dimensional representation space.

We use the implementation of RotNet released by its authors[4]. Note that the original RotNet is applied to large image datasets and has a deep network architecture, involving four convolutional blocks with three convolutional layers for each block. We found directly using the original architecture is too deep for *Olivetti* and performs ineffectively as the data contains only 400 image samples. Therefore, we simplify the architecture of RotNet and derive four variants of RotNet, including RotNet$_{4\times2}$, RotNet$_{4\times1}$, RotNet$_{3\times1}$ and RotNet$_{2\times1}$. Here RotNet$_{a\times b}$ represents RotNet with $a$ convolutional blocks and $b$ convolutional layers for each block. Note that RotNet$_{2\times1}$ is the simplest variant we can derive that works effectively. We evaluate the original RotNet, its four variants and the combination of these five RotNets and RDP.

Table 13: NMI and F-score performance of K-means clustering using RDP, RotNet, and RotNet+RDP based feature representations of the image dataset Olivetti.

|  | NMI Performance | F-score Performance |
| --- | --- | --- |
| Org | $0.778 \pm 0.014$ | $0.590 \pm 0.029$ |
| RDP | $\mathbf{0.805 \pm 0.012}$ | $\mathbf{0.638 \pm 0.026}$ |
| RotNet | $0.467 \pm 0.014$ | $\mathbf{0.243 \pm 0.014}$ |
| RotNet+RDP | $\mathbf{0.472 \pm 0.011}$ | $0.242 \pm 0.011$ |
| RotNet$_{4\times2}$ | $\mathbf{0.518 \pm 0.010}$ | $0.281 \pm 0.014$ |
| RotNet$_{4\times2}$+RDP | $0.517 \pm 0.010$ | $\mathbf{0.282 \pm 0.014}$ |
| RotNet$_{4\times1}$ | $0.519 \pm 0.010$ | $0.283 \pm 0.014$ |
| RotNet$_{4\times1}$+RDP | $\mathbf{0.536 \pm 0.010}$ | $\mathbf{0.298 \pm 0.011}$ |
| RotNet$_{3\times1}$ | $0.526 \pm 0.014$ | $0.303 \pm 0.018$ |
| RotNet$_{3\times1}$+RDP | $\mathbf{0.567 \pm 0.010}$ | $\mathbf{0.336 \pm 0.015}$ |
| RotNet$_{2\times1}$ | $0.561 \pm 0.010$ | $0.339 \pm 0.016$ |
| RotNet$_{2\times1}$+RDP | $\mathbf{0.587 \pm 0.009}$ | $\mathbf{0.374 \pm 0.015}$ |

The evaluation results are presented in Table 13. Impressively, RDP can significantly outperform RotNet and all its four variants on *Olivetti*. It is interesting that Org (i.e., performing K-means clustering on the original $64 \times 64$ vector space) also obtains a similar superiority over the RotNet family. This may be because *Olivetti* is too small to provide sufficient training samples for RotNet and its variants to learn its underlying semantic abstractions. This conjecture can also explain the increasing performance of RotNet variants with decreasing complexity of the RotNet architecture. Similar to the results on the raw text data, applying RDP on the RotNet projected spaces can also learn substantially more expressive representations than the representations yielded by RotNet and its variants, especially when the RotNet methods work well, such as the two cases: RotNet$_{3\times1}$ vs. RotNet$_{3\times1}$+RDP and RotNet$_{2\times1}$ vs. RotNet$_{2\times1}$+RDP.

## H   Performance Evaluation in Classification

We also performed some preliminary evaluation of the learned representations in classification tasks using a feed-forward three-layer neural network model as the classifier. We used the same datasets as in the clustering task. Specifically, the representation learning model first outputs the new representations of the input data, and then the classifier performs classification on the learned representations. RDP is compared with the same competing methods HLLE, SRP, AE and COP as in clustering. F-score is used as the performance evaluation metric here.

---

[4]The released code of RotNet is available at https://github.com/gidariss/FeatureLearningRotNet.

The results are shown in Table 14. Similar to the performance in clustering and anomaly detection, our model using only the random distance prediction loss $L_{rdp}$, i.e., RDP$\setminus L_{aux}^{clu}$, performs very favourably and stably on all the five datasets. The incorporation of $\setminus L_{aux}^{clu}$ into the model, i.e., RDP, helps gain some extra performance improvement on datasets like *20news*, but it may also slightly downgrade the performance on other datasets. An extra hyperparameter may be added to control the importance of these two losses.

Table 14: F-score performance of classification on five real-world datasets.

| Data | HLLE | SRP | AE | COP | RDP$\setminus L_{aux}^{clu}$ | RDP |
|---|---|---|---|---|---|---|
| **R8** | 0.246 | 0.895 | 0.874 | 0.860 | 0.900 | **0.906** |
| **20news** | 0.005 | 0.733 | 0.709 | 0.718 | 0.735 | **0.753** |
| **Olivetti** | 0.895 | 0.899 | 0.820 | 0.828 | **0.900** | 0.896 |
| **Sector** | 0.037 | 0.671 | 0.645 | 0.689 | 0.690 | **0.696** |
| **RCV1** | 0.766 | 0.919 | 0.918 | N/A | **0.940** | 0.926 |

