# OpenReview forum: "Unsupervised Representation Learning by Predicting Random Distances"
_ICLR.cc/2020/Conference — Reject_

### Official Review · AnonReviewer2 · 2019-10-21
**Official Blind Review #2**

**Rating:** 6

**Review:**

The authors discuss the a novel technique, called Random Distance Prediction, to learn rich features from domains where massive data are hard to produce; in particular, they focus on the two tasks of anomaly detection and clustering.
The paper is well written and understandable by a non-specialistic audience; introduction and references are adequate, and the theoretical analysis is reasonably explained, although the two optional losses should have been discussed more deeply.
Results are fairly supporting the authors' claim: however, the improvement in performances w.r.t. alternative approaches are limited in most cases, and the contribute of the optional losses is somehow unclear and inconsistent across different datasets. It would be interesting also to check how stable is the method (i.e. the losses) for different underlying network architectures, and also how RDP compares to very basic approaches employing dimensionality reduction algorithms such as t-SNE or UMAP.

**Experience Assessment:**

I have read many papers in this area.

**Review Assessment: Checking Correctness Of Derivations And Theory:**

I assessed the sensibility of the derivations and theory.

**Review Assessment: Checking Correctness Of Experiments:**

I assessed the sensibility of the experiments.

**Review Assessment: Thoroughness In Paper Reading:**

I read the paper thoroughly.

---

> ### Author Response · Authors · 2019-11-15
> **Concerns  addressed (the optional losses, network architectures, new comparison)**
>
> Thanks for your positive and constructive comments, which helps substantially refine our paper. Your concerns are addressed as follows.
>
> 1. The discussion of two optional losses. In the refined paper, we have created the Section 2.2. to focus on the discussion of the two optional losses, and Sections 1 and 2.1 have been substantially refined to discuss the relationship between the proposed random distance prediction loss and these two optional losses. Hope the paper better explains the optional losses now.
>
> 2. The improvement in performance. In a few cases of the clustering datasets, it is true that the improvement of RDP is marginal w.r.t. Org in terms of NMI, but the improvement is very substantial in terms of F-score on datasets such as R8, Olivetti and RCV1; on most clustering datasets, RDP achieves substantial improvement over all the other four competing methods in both performance metrics. In the anomaly detection datasets, our method RDP outperforms all the five competing methods in both AUC-ROC and AUC-PR in at least 12 out of 14 datasets. This improvement is statistically significant at the 95% confidence level according to the two-tailed sign test across the 14 datasets.
>
> 3. The contribution of the optional losses. In general, the optional auxiliary losses are designed to provide complementary supervision information for the random distance prediction loss, resulting in different learning constraints. These complementary supervision work in most cases. For example, in the anomaly detection task, as shown in Table 3, in 13 out of 14 datasets, RDP that uses the random distance prediction loss and the optional novelty loss performs better than the ablated RDP versions, RDP\Lrdp and RDP\Laux, which remove either the random distance prediction loss or the optional novelty loss; similarly, in at least 4 out of 5 clustering datasets, RDP that uses the random distance prediction loss and the optional reconstruction loss performs better than the ablated RDP versions, RDP\Lrdp and RDP\Laux, which removes either the random distance prediction loss or the optional novelty loss. Therefore, the optional losses generally have an important contribution to further improve RDP.
>
> 4. The sensitivity of RDP w.r.t. different underlying network architectures. We have added additional experimental results to show the sensitivity of RDP w.r.t. different network architectures in Figures 2-4 in Appendix F. We performed the sensitivity test by varying a key component of the network, the number of units in the feature learning layer (i.e., the representation dimension in the new space). Our results show that RDP performs stably with a wide range of representation dimension options in both anomaly detection and clustering tasks.
>
> 5. Comparison to t-SNE and UMAP. The t-SNE and UMAP methods are very sensitive to hyperparameters. Some best results using UMAP in clustering are provided in Table 1 below. t-SNE works less effectively than t-SNE. Similar to t-SNE and UMAP, HLLE is also one popular manifold learning methods for dimension reduction. We empirically found that HLLE works generally better than t-SNE and UMAP in our experiments. So, we reported the results of HLLE in the paper only. We believe that t-SNE and UMAP work best in preserving proximity information in extremely lower (e.g., 2-D or 3-D) dimensional space, so they are more popular in data visualisation.
>
> Table 1: F-score performance of K-means on the UMAP, HLLE and RDP projected spaces.
> +---------------------------------------------------------------------------+
> | Data     |        UMAP      |         HLLE        |          RDP         |
> +---------------------------------------------------------------------------+
> | R8         |  0.085 ± 0.000| 0.085 ± 0.000 | 0.360 ± 0.055   |
> +---------------------------------------------------------------------------+
> | 20news| 0.054 ± 0.003 | 0.007 ± 0.000 | 0.119 ± 0.006   |
> +---------------------------------------------------------------------------+
> | Olivetti |  0.164 ± 0.009| 0.684 ± 0.024 | 0.638 ± 0.026   |
> +---------------------------------------------------------------------------+
> | Sector  | 0.024 ± 0.001 | 0.062 ± 0.001 | 0.191 ± 0.007   |
> +---------------------------------------------------------------------------+
> | RCV1    | 0.341 ± 0.000 | 0.342 ± 0.000 | 0.572 ± 0.003   |
> +---------------------------------------------------------------------------+

---

### Official Review · AnonReviewer1 · 2019-10-21
**Official Blind Review #1**

**Rating:** 3

**Review:**

This paper proposed a method of unsupervised representation by transforming a set of data points into another space while maintaining the pairwise distance as good as possible. The paper is well structured with background literatures, formula, as well as experiments to show the advantage of the proposed method. I find it generally interesting, with the following major concerns.

1. Representation or dimension reduction? If the original space is a structured space like Euclidean space, then effectively this paper's method coincides with regular distance preserving method in dimension reduction, and Johnson-Lindenstrauss theories. If the original space is not structured or doesn't naturally have a good distance measure, then the proposed method cannot work. For example, if the original dataset is a set of documents, and the task is to do representation learning to convert each document into a compact vector. However, there's no good distance metric for the document space. If TF-IDF is used, then the representation space also inherits TF-IDF type features which is not desired. If more advanced similarity is used for the document space, then the role of representation learning is not essential anymore as that similarity measure can already help the downstream tasks.

2. Section 3 the theoretical analysis. This part seems like a collection of previous works and contains minimal information about the proposed method.

3. Some writing issues, like page 4 line 7 about the equation numbering.

**Experience Assessment:**

I have published one or two papers in this area.

**Review Assessment: Checking Correctness Of Derivations And Theory:**

I assessed the sensibility of the derivations and theory.

**Review Assessment: Checking Correctness Of Experiments:**

I assessed the sensibility of the experiments.

**Review Assessment: Thoroughness In Paper Reading:**

I read the paper at least twice and used my best judgement in assessing the paper.

---

> ### Author Response · Authors · 2019-11-15
> **Concerns addressed (relation to regular distance preserving methods, theoretical analysis, typos)**
>
> Thank you for your constructive comments that help substantially improve our paper. Your concerns are addressed as follows.
>
> 1.1: Our method actually takes one significant step further w.r.t. many regular distance preserving methods. Regular methods well preserve a large amount of local proximity information, but also often preserve misleading proximity when their underlying assumption is inexact for a given dataset. By minimising the difference between the predicted distance and the predefined distance yielded by the regular distance preserving method (e.g., random projection), our method essentially leverages the preserved data proximity and the power of neural networks to learn globally consistent local proximity (e.g., the genuine proximity information) and rectify the inconsistent proximity information (e.g., the inaccurate ones due to the inexact assumption) in a new space. Therefore, our method is equivalent to optimise the given proximity information using imperfect supervision information.  Our method learns a significantly improved feature space out of the original distance preserving space when the genuine proximity information provided by the regular distance preserving method is sufficient. Therefore, our method is built upon regular distance preserving methods to learn better optimised proximity information for more expressive feature representations. This is the key driving force to enable our method to achieve substantially better performance than the counterparts that work on the original data space or the regular distance preserving methods-based spaces. We have rewritten the introduction and Section 3.3 to highlight this point.
>
> 1.2: Yes, it is true that our method RDP requires distance information as the supervision to perform the feature learning, but, as demonstrated in the clustering experiments where text and image datasets are used, our method can work very effectively in handling text/image data. In our paper we use simple data transformation (such as bag-of-words model for text data and treating each pixel as a feature unit for image data) to convert the raw text/image data into feature vectors, and then use our random distance prediction method to perform the feature learning. Our extensive results show that this simple transformation can enable our method to effectively handle raw data that is not structured in nature.
>
> How is the performance of RDP compared to those advanced representation learning methods that are specifically designed for raw text/image datasets? This is the missing part of our paper. So, we perform an empirical comparison between RDP and these advanced representation learning methods to answer this question. The results are now added in Tables 12 and 13 in Appendix G in the refined version of our paper. We brief our empirical findings as follows. In general, the advanced representation methods Doc2Vec (Le & Mikolov, ICML 2014) and RotNet (Gidaris et al., ICLR 2018)  are respectively used as the competing method for learning representations of raw texts and images. Our empirical results show that RDP with simple transformation can perform very comparable to,  or substantially better than, Doc2Vec and RotNet. Also, we derive Doc2Vec+RPD (RotNet+RDP) that works on the dense vectors yielded by Doc2Vec (RotNet). Our results show that Doc2Vec+RPD (RotNet+RDP) can also achieve significant improvement over Doc2Vec (RotNet). Doc2Vec and RotNet may perform better if they are properly pretrained. In such cases, Doc2Vec+RPD (RotNet+RDP) may obtain further improvement, as we demonstrated in Tables 12 and 13. Therefore, we believe that RDP can also be an important approach to learn representations of those raw data.
>
> 2. The theoretical analysis in Sections 3.1 and 3.2 is to show the proven distance preserving properties of random projection. We agree that, as we stated in the paper, the used properties are indeed from the previous works, for which we put clear references and statements. However, these proven properties provide critical theoretical foundation to the proposed method. We have refined Section 3 to highlight this point. Particularly, Since the random distance information is used as the supervisory signal in our proposed method RDP, our method can work only if these random distances need to preserve original proximity information. The proven properties in Sections 3.1 and 3.2 show that either linear or non-linear random projection methods can be used to obtain such preserved distances efficiently. So, they provide strong theoretical motivation and proven support for the use of random distances as a reliable supervisory signal in RDP. In Section 3.3, we unfold RDP and link it to supervised learning with imperfect supervision information, which provides an aspect for understanding why RDP can learn better representation space out of its input space.
>
> 3. Thank you for pointing out the typo. We have fixed this typo and a number of other writing issues.

---

### Official Review · AnonReviewer4 · 2019-11-08
**Official Blind Review #4**

**Rating:** 6

**Review:**


###### Overall Recommendation
I vote for the “Weak Accept” decision for this paper.

### Summary
This paper introduces a novel model termed Random Distance Prediction model, it can predict data distances in randomly projected space. The distance in projected space is used as the supervisory signal to learn representations without any manually labeled data, avoiding the concentration and inefficiency problem when dealing with high-dimensional tabular data. Their main contribution is extending the random distance in projected spaces to approximate the original distance information of the hight-dimensional tabular data effectively. Overall, the idea in this paper is interesting and effective, the experiment results in two typical unsupervised tasks (anomaly detection and clustering) also look very promising. However, the writing sometimes has unclear descriptions, given these clarifications in an author's response, I would be willing to increase the score.

### Strengths
1. The illustration of the authors' idea is clear and concise.
2. The theoretical analysis of the proposed method is solid and systematic, the validity of the subparts have been proven previously.
3. The experiment part is well organized. The RDP model are compared with several state-of-the-art unsupervised learning methods in 19 real-world datasets of various domains. The experimental setup is solid with realistic considerations, the results are very convincing and promising.
4. This paper provides sufficient detail for reproducing the results.

### Weaknesses
Lack of systematic description of the authors’ major contribution. The proposed model looks more like a combination of previous conclusions, which makes readers feel the core parts of this paper build heavily on previous work.

### Questions
1. What is the relation between “random distance prediction loss” and “task-dependent auxiliary loss”?
2. Are there any solutions and references about how to choose the task-dependent loss L_aux?
3. Why you shade the second part of the loss function in Figure 1;
4. How long is it for training the proposed model and getting the experiment results? Does the RDP model still outperform the other algorithms?

### Suggestions to improve the paper
1.  It would be better to reorganize Section 1 and Section 2, please describe the contribution in a more systematic way.
2. Add details for the architecture of the model, please give more descriptions about Figure 1.
3. It might also help to add an algorithm comparison box for the test time for the proposed method.

### Minor Edit Suggestions
1. It would be better to give more descriptions about Figure 1; the lower right part in Figure 1 is not explained in the caption; the shadow part in Figure 1 is not precise.
2. Figure 1 was bad organized, please make the legend readable size.
3. I don't think there exist the proofs of Eqns. (2)-(4) in the reference paper (Vempala, 1998), which was written in Page 4. The number should be revised.

**Experience Assessment:**

I have read many papers in this area.

**Review Assessment: Checking Correctness Of Derivations And Theory:**

I assessed the sensibility of the derivations and theory.

**Review Assessment: Checking Correctness Of Experiments:**

I assessed the sensibility of the experiments.

**Review Assessment: Thoroughness In Paper Reading:**

I read the paper thoroughly.

---

> ### Author Response · Authors · 2019-11-15
> **Concerns addressed (contribution statement, relation between the main and auxiliary losses, computational efficiency, selection of auxiliary loss, framework description, typos)**
>
> Thank you for the positive and constructive comments, which helps substantially enhance the paper. We addressed your concerns/suggestions as follows.
>
> 1. Systematic description of the authors’ major contribution. We have rewritten the last part of Section 1 to provide a more systematic description of our main contributions in this work. The three main contributions are now read as: (1) We propose a random distance prediction formulation, which is very simple yet offers a highly effective supervisory signal for learning expressive feature representations that optimise the distance preserving in random projection. The learned features are sufficiently generic and work well in enabling different downstream learning tasks. (2) Our formulation is flexible to incorporate task-dependent auxiliary losses that are complementary to random distance prediction to further enhance the learned features, i.e., features that are specifically optimised for a downstream task while at the same time preserving the generic proximity as much as possible. (3) As a result, we show that our instantiated model termed RDP enables substantially better performance than state-of-the-art competing methods in two key unsupervised tasks, anomaly detection and clustering, on 19 real-world high-dimensional tabular datasets.
>
> We also reorganised Section 2 into two subsections, with one subsection to introduce the proposed random distance prediction idea and another subsection to introduce the incorporation of the optional loss. This is aligned to our stated contributions.
>
> 2. The relation between “random distance prediction loss” and “task-dependent auxiliary loss”. The task-dependent auxiliary loss provides complementary information to the random distance prediction loss. Specifically, random distance prediction optimises the preserved local proximity information, while the task-dependent auxiliary losses, including reconstruction loss and the novelty loss, learns global features that are important to a given unsupervised task. We explicitly highlight this complementary relation in both Sections 1 and 2 in the refined paper.
>
> 3. Solutions and references about how to choose the task-dependent loss L_aux. We didn’t find useful references for guiding the choice of the task-dependent loss. Our intuition of choosing these auxiliary losses are as follows. The reconstruction loss is an effective loss to learn globally consistent features which are generally important to clustering; the novelty loss is devised to learn the frequency of underlying patterns in the data, so it is critical to anomaly detection as anomalies often correspond to rare events. It is interesting and of great importance in real-world applications if we could have some methods to learn how to choose auxiliary losses to further enhance particular models. We will study this problem in our future work.
>
> 4. Several problems in Figure 1: more description; shadow part; legend size. We have rewritten the caption of Figure 1 to provide sufficient description for readers to have a quick sense of the insight and process of our model. The previous shadow part was to highlight the random distance. We now remove the shadow to avoid distraction. The figure in the lower right position is now resized with readable axes and legend. We also provide a description in the figure caption to explain how the figure was created.
>
> 5. Computational efficiency of the proposed method. The training of our method is generally comparably fast to the two recently proposed deep methods REPEN and DAGMM. For example in the high-dimensional and large-scale anomaly detection dataset Census, our method takes 3780 seconds, REPEN takes 3093 seconds, DAGMM takes 7891 seconds, and AE and RND take about 1400 seconds. So, the training efficiency of our method is around the middle among the competing deep methods. The traditional method iForest generally has better efficiency, e.g., it does not involve any optimisation and takes about 155 seconds to build their model on Census. Nevertheless, since training time can vary significantly using different training strategies in deep learning-based methods, it is difficult to have a fair comparison of the training time. Moreover, the models can often be trained offline. Thus, we focus on comparing the runtime at the testing stage, which is more important in real-world applications. The comparison of the testing runtime for both tasks is provided in Tables 10 and 11 in Appendix F. Our results show that our method is generally comparably fast to the most efficient competing method.
>
> 6. Typos in Page 4. We have fixed the typos.

---

### Author Response · Authors · 2019-11-15
**Revision summary**

Dear All reviewers,

Tremendous thanks for your constructive comments that have helped us substantially enhance the paper. In summary, we have made the following refinement in the updated paper.

1. We have rewritten the last part of the introduction and reorganised the second section to better highlight the main contributions of this work.

2. We have substantially refined Section 3 to allow readers to have a more straightforward understanding of the theoretical foundation of our method, and to understand the underlying optimisation our method does

3. A series of new empirical results are added in Appendix E to G to address several concerns raised. Specifically, in Appendix E, our results show that our method performs stably with different representation dimensions in both unsupervised tasks. In Appendix F, we show empirical results for the testing runtime, which demonstrate that our method is generally comparably fast to the most efficient methods in both tasks. In Appendix G, we make a comparison between our method and advanced representation learning methods that are specifically designed for raw text/image data. The results show that our method can outperform these advanced representation learning methods in most cases.

4. All the minor issues have been fixed in the updated paper.

Please refer to the updated paper for the detailed revision. We believe the revised paper have addressed all your major concerns. Please see our replies below for the detailed responses to your comments. Thanks.

Best regards,
Authors of Paper 1545

---

### Decision · Program_Chairs · 2019-12-19

**Decision:**

Reject

**Comment:**

The reviewers agree that this is an interesting paper but it required major modifications. After rebuttal, thee paper is much improved but unfortunately not above the bar yet. We encourage the authors to iterate on this work again.